# Association between Smoking and Periodontal Disease in South Korean Adults

**DOI:** 10.3390/ijerph20054423

**Published:** 2023-03-01

**Authors:** Ka-Yun Sim, Yun Seo Jang, Ye Seul Jang, Nataliya Nerobkova, Eun-Cheol Park

**Affiliations:** 1Department of Public Health, Graduate School, Yonsei University, Seoul 03722, Republic of Korea; 2Institute of Health Services Research, Yonsei University, Seoul 03722, Republic of Korea; 3Department of Preventive Medicine, Yonsei University College of Medicine, Seoul 03722, Republic of Korea

**Keywords:** periodontal health, periodontal disease, smoking, Community Periodontal Index (CPI)

## Abstract

Smoking poses a threat to global public health. This study analyzed data from the 2016–2018 National Health and Nutrition Examination Survey to investigate smoking’s impact on periodontal health and identify potential risk factors associated with poor periodontal health in Korean adults. The final study population was 9178 patients, with 4161 men and 5017 women. The dependent variable was the Community Periodontal Index (CPI), to investigate periodontal disease risks. Smoking was the independent variable and was divided into three groups. The chi-squared test and multivariable logistic regression analyses were used in this study. Current smokers had a higher risk of periodontal disease than non-smokers (males OR: 1.78, 95% CIs = 1.43–2.23, females OR: 1.44, 95% CIs = 1.04–1.99). Age, educational level, and dental checkups affected periodontal disease. Men with a higher number of pack years had a higher risk of periodontal disease than non-smokers (OR: 1.84, 95% CIs = 1.38–2.47). Men who quit smoking for less than five years had a higher risk of periodontal disease than non-smokers but lower than current smokers (current OR: 1.78, 95% CIs = 1.43–2.23, ex OR: 1.42, 95% CIs = 1.04–1.96). Those who had quit smoking for less than five years had a higher risk of periodontal disease than non-smokers but lower than current smokers (males OR: 1.42, 95% CIs = 1.04–1.96, females OR: 1.11, 95% CIs = 1.71–1.74). It is necessary to motivate smokers by educating them on the importance of early smoking cessation.

## 1. Introduction

Smoking is one of the biggest threats to public health [1]. According to the World Health Organization, more than 8 million people have been killed, including approximately 1.2 million deaths from exposure to secondhand smoke [2]. Moreover, since tobacco has more than 7000 toxic chemicals [3], smoking is associated with numerous preventable chronic diseases [4]. In Korea, the smoking rate has been decreasing; however, as of 2018, the prevalence of daily smoking among men in Korea reached 30.5%, the third-highest rate among the Organization for Economic Co-operation and Development (OECD) members [5]. The authorities have made intensive efforts to eliminate tobacco use by implementing strong and effective tobacco control policies and measures, such as cigarette tax hikes and media campaigns [4,6].

The association between smoking and various diseases, including major causes of death, has been well-established. A cohort study in the US reported that smokers had a higher risk of developing bladder cancer and pancreatic cancer than non-smokers [7]. Another study found that smokers were more likely to have elevated levels of blood insulin and triglycerides compared to non-smokers [8,9].

Smoking can negatively impact the oral cavity, particularly in non-inflammatory oral diseases [10]. Harmful substances in tobacco products, such as nicotine, can harm the gingival tissue, decrease blood flow to the gums, and compromise the immune system [11]. Tobacco use can increase susceptibility to oral infections, stain teeth, cause dryness in the mouth, and delay the healing of oral wounds [12].

Periodontal diseases are considered to be chronic destructive inflammatory diseases [13]. They are characterized by the destruction of the periodontal tissue, loss of adhesion to connective tissues, loss of alveolar bone, and the formation of pathological sacs around the teeth [14,15,16]. In addition, poor periodontal health is associated with systemic diseases, such as cancer, heart disease, and diabetes; therefore, management is important [17,18,19]. Previous studies have shown that smoking is associated with poor periodontal health, even among young adults [20]. Another study in Korea revealed that quitting smoking within a decade could potentially improve periodontal health for former smokers [21]. A study in the US, which used large-scale data, concluded that smoking is a significant risk factor for periodontitis and may account for more than 50% of periodontitis in adults [22].

While previous studies have examined the association between smoking and periodontal diseases, additional evidence is needed to encourage healthy habits that promote smoking cessation. This study aimed to investigate the relationship between smoking and the risk of periodontal diseases in Korean adults, using a nationwide cross-sectional survey with a large sample size. Furthermore, this study aimed to provide more robust evidence for the importance of early smoking cessation by analyzing the relationship between smoking cessation in five-year intervals, which was more detailed than in previous studies.

## 2. Materials and Methods

### 2.1. Data

The data for this study were obtained from the 2016–2018 Korea National Health and Nutrition Examination Survey (KNHANES) and used a separate raw dataset (HNYN_OE). The KNHANES has been conducted by the Korea Disease Control and Prevention Agency (KDCA) since 1998 to investigate national statistics through a survey of the health level, health-related behavior, and nutritional status of 10,000 Koreans annually. The KDCA Research Ethics Review Board approved the data collection protocols for the KNHANES. The data are available for download from the KDCA website (https://knhanes.kdca.go.kr/knhanes/sub03/sub03_02_05.do, accessed on 1 January 2023). Thus, this study did not need extra approval from the ethics review board. The KNHANES is a self-reported survey using a stratified, two-stage, clustered sampling design conducted annually for South Koreans of all ages, divided into three age groups: (children: 1–11 years old, adolescents: 12–18 years old, and adults: 19 years or older).

### 2.2. Study Population

The total number of participants who completed the health examination survey for KNHANES 2016–2018 was 16,489 (7485 males and 9004 females). The exclusion criteria consisted of three categories: (a) under 19 years of age (N = 3299), (b) unable to perform oral examination due to tooth loss (N = 2581), and (c) missing values in health assessment or survey (N = 1440). The final study population was 9178, with 4161 men and 5017 women (Figure 1).

### 2.3. Variables

The dependent variable in this study was the Community Periodontal Index (CPI), used to measure the risk of periodontal disease. The oral health examinations were conducted by public health dentists and local public health dentists at the city and provincial levels under the supervision of the Korea Disease Control and Prevention Agency (KDCA). The risk to periodontal health was assessed by dividing the upper and lower jaws into three sections and recording the highest CPI score for each section. The CPI score was based on periodontal pocket depth, calculus attachment, and gingival bleeding measurements. The scores ranged from 0 to 4, with 0 indicating healthy, 1 indicating bleeding, 2 indicating dental calculus, 3 indicating a superficial periodontal pocket of 4–5 mm, and 4 indicating a deep periodontal pocket of 6 mm or more. Using the sum of the CPI scores, we assessed the risk of periodontal disease as the outcome variable.

The independent variable was the smoking status, classified into three groups: non-smokers, ex-smokers, and current smokers. Smoking status was based on the question, “Do you currently smoke cigarettes?”. We also used pack years and smoking cessation status as variables in the subgroup analysis. Pack years indicate the number of cigarettes a person has smoked in their lifetime, calculated by multiplying the total number of cigarettes smoked per day by the total number of years a person smoked.

The covariate variables were controlled for, as potential confounding factors. These included socioeconomic factors, such as sex, age, household income, and region, and factors related to health behaviors, such as current drinking status and physical activity. Oral health habits were also included as covariates. Teeth brushing frequency was investigated, based on the number of times teeth were brushed during the previous day, while dental checkup status was surveyed based on the question, “Did you have a dental checkup in the past 12 months?”.

### 2.4. Statistical Analysis

A chi-squared test was conducted to explore the general characteristics of the study population. The general characteristics of the final study population were represented as frequency and percentage. To assess the relationship between smoking and periodontal disease using the sum of the CPI scores in adults, we used multivariable logistic regression analysis with covariate adjustment. Subgroup analyses were performed to evaluate the relationship between pack years, smoking cessation status, and periodontal disease. All the results were presented as odds ratios (ORs) and 95% confidence intervals (CIs). The analyses were performed using stratified sampling variables. All the estimates were estimated using weighted variables to generalize the data. SAS version 9.4 software (SAS Institute, Cary, NC, USA) was used for all the statistical analyses. Statistical significance was determined as a two-sided *p*-value of <0.05.

## 3. Results

Table 1 summarizes the characteristics of the study population, classified according to sex. Of the 9178 participants, 4161 were male (45.3%), and 5017 were female (54.7%). A total of 3042 (73.1%) males and 3143 (62.6%) females had periodontal disease risks, as expressed by the CPI. Among the males, 1484 (35.7%) were current smokers, 1623 (39.0%) were ex-smokers, and 1054 (25.3%) were non-smokers. Among the females, 282 (5.6%) were current smokers, 344 (6.9%) were ex-smokers, and 4391 (87.5%) were non-smokers.

Table 2 presents the multivariate logistic regression analysis results that explore the association between smoking and periodontal disease while adjusting for covariates. The smokers had a higher risk of periodontal disease than the non-smokers. While the ex-smokers were statistically insignificant, the current smokers were significant for males (OR: 1.78, 95% CIs = 1.43–2.23) and females (OR: 1.44, 95% CIs = 1.04–1.99). As age increased, the participants showed an elevated risk of periodontal disease. The participants with a middle school education or lower had a higher risk of periodontal disease than those with a college education (males OR: 1.63, 95% CIs = 1.15–2.23, females OR: 1.59, 95% CIs = 1.18–2.14). The individuals who did not receive dental checkups were likelier to have periodontal diseases (males OR: 1.62, 95% CIs = 1.36–1.93, females OR: 1.60, 95% CIs = 1.39–1.84).

Table 3 presents the results of the subgroup analysis for the independent variables stratified by smoking behavior. Most of the ex-smokers did not have significant results. The observed results were more significant in the males than in the females. The risk of periodontal disease generally increased with age in men who are current smokers but was not statistically significant in their 50s. The current smokers had a higher risk of periodontal disease in all education levels, and the risk was highest for those with middle school education or lower (OR: 3.15; 95% CIs = 1.37–7.21). The current smokers had a risk of periodontal disease, regardless of their physical activity status (male, adequate: OR = 1.83, 95% CIs = 1.35–2.50; inadequate: OR = 1.77, 95% CIs = 1.30–2.42). Similarly, regardless of whether they received regular dental checkups, current smokers had a higher risk of periodontal disease (male, checkups: OR = 1.90, 95% CIs = 1.40–2.59; no checkups: OR = 1.73, 95% CIs = 1.29–2.33).

The results of the subgroup analysis, which were stratified by pack years and smoking cessation, are presented in Table 4. The males showed a statistically significant positive association. Those with a higher number of pack years had a higher risk of periodontal diseases than the non-smokers (over 20 pack years OR: 1.84, 95% CIs = 1.38–2.47). Those who had quit smoking for less than five years had a higher risk of periodontal disease than the non-smokers but lower than the current smokers (males OR: 1.42, 95% CIs = 1.04–1.96; females OR: 1.11, 95% CIs = 1.71–1.74).

## 4. Discussion

Despite the reduction in smoking prevalence over the past 30 years, the total number of smokers has increased from 0.99 billion in 1990 to 1.14 billion in 2019 worldwide, due to population growth [23]. The American Academy of Periodontology has pointed out that smoking negatively impacts the healing and treatment of periodontitis [24]. The purpose of the study was two main issues. First, we used a nationwide survey with a large sample size to investigate the association between smoking and periodontal disease. Second, we attempted to support the importance of early smoking cessation by analyzing the relationship between smoking cessation in five-year intervals compared to the previous studies using ten-year intervals.

The mechanisms underlying the association between smoking and periodontal disease were the following. Smoking stimulates the establishment of pathogenic microflora, diminishes the immune host response, and elevates the release of inflammatory mediators [14,15,16,25]. As smokers are more likely to absorb pathogenic microorganisms than non-smokers, previous studies have reported an increase in particular pathogens in smokers, such as *Actinobacillus actinomycetemcomitans* and *Bacteroides forsythus*, although the pathogen levels may have varied, based on the methods used in the studies [14,26,27]. Smoking can affect host inflammatory and immune responses, such as the immunosuppressive effects of macrophages on cell-mediated immune responses, inhibition of human periodontal ligament fibroblast migration, and repression of alkaline phosphatase production by nicotine [28,29]. As with this mechanism, Table 2 shows current smokers had a higher risk of periodontal diseases than non-smokers. It supports previous studies’ results that smoking is a risk factor for oral health, even among young smokers [20]. Additionally, the results in Table 2 are in the same vein as previous studies, showing that smoking significantly influences periodontitis, using a large sample in the US [22].

Notable points in Table 2 were the results of age, education level, and dental checkup status variables. In the case of age, it was consistent with the results of previous studies that the prevalence of periodontal disease tends to increase as the age of participants increases. Previous studies in Brazil and India have reported that age increases affect the severity and prevalence of periodontal disease, regardless of gender [30,31]. The education level affected periodontal health. Middle school or lower education participants had a higher risk of periodontal disease than those with a college or higher education. As some studies have reported similar findings [22,32], education progressively decreases the risk of periodontal diseases. This finding implies that education regarding periodontal health is important. There was a higher risk of periodontal disease in people who did not undergo oral examinations, which supports previous studies that those who regularly underwent oral examinations had a lower risk of periodontal disease than those who did not [33,34].

In Table 3, male smokers in their 50s were not statistically significant. This counterintuitive finding can be explained by aging, which affects tooth loss [4,35]. Even if good physical activity habits and dental checkups were regularly undertaken, the current smokers had a higher risk of periodontal disease than the non-smokers. This result could explain that current smokers cannot avoid the risk of periodontal disease, even if they have good health habits.

As shown in Table 4, the men with high pack years had a higher risk of periodontal disease. However, the women showed statistically insignificant results. The results can be explained in the WHO Framework Convention on Tobacco Control (WHO FCTC) context. The WHO FCTC emphasized the need to consider gender when developing tobacco control strategies, as perceptions of smoking habits related to gender continue to differ depending on social contexts and cultural norms. Specifically, in Confucian Asian countries, there is still a tendency for views on female smoking to be more conservative than those on male smoking [36]. The smoking rate of women is also increasing in Korea. However, considering the social context, the data on the female smoking rate collected by voluntary reporting may not be accurate, due to the opposing views of some female smokers.

We found that ex-smokers with relatively short smoking cessation periods had a lower risk of periodontal disease than current smokers. This result can be compared to a previous study that reported the possibility of reversing the risk of periodontal disease if an individual quits smoking for ten years [15]. The sentence that smoking is harmful was too clear and simple, but the results of this study tried to support the importance of early smoking cessation. It could motivate smokers to quit smoking by revealing that people who quit smoking for a relatively short period, fewer than five years, have a lower risk of periodontal disease.

This study had several limitations. First, clarifying an inverse causal association was difficult since it was a cross-sectional study. Second, the KNHANES data were collected through a self-reported survey. The data on smoking behavior, health habits, and socioeconomic variables may not be accurately estimated. There was a possibility of recall bias. Third, it was impossible to identify the type of smoking, such as whether the participants used conventional cigarettes, e-cigarettes, or both. In addition, we could not use biological indicators, such as urine cotinine, in the subjects. Therefore, further studies are needed, considering these limitations.

Despite these limitations, our study has several strengths. The main strength of this study is the use of nationally representative, large, and high-quality data. KNHANES was conducted using a random cluster design, which can generalize the study’s results to the general population. Second, oral health examination datasets collected by public health doctors may effectively estimate periodontal disease risks. It was possible to estimate the risk of periodontal disease more precisely by using the CPI score through a doctor’s examination than by using the participant’s subjective oral symptom self-reported survey. Third, our study supported the importance of early smoking cessation. Compared to previous studies, we provided more proactive evidence for the importance of early smoking cessation.

## 5. Conclusions

This study demonstrated a strong association between smoking and periodontal disease in South Korean adults. Long-term smoking was closely related to poor periodontal health. The findings that even a relatively short period of smoking cessation, less than five years, had a positive impact on periodontal disease could be a powerful motivator for smokers. There is a need for effective tobacco control measures to reduce the prevalence of periodontitis. Tailored smoking cessation policies and educational interventions, highlighting the benefits of short-term smoking abstinence in reducing periodontal disease risk, could encourage current smokers to quit and ultimately improve public oral health.

## Figures and Tables

**Figure 1 ijerph-20-04423-f001:**
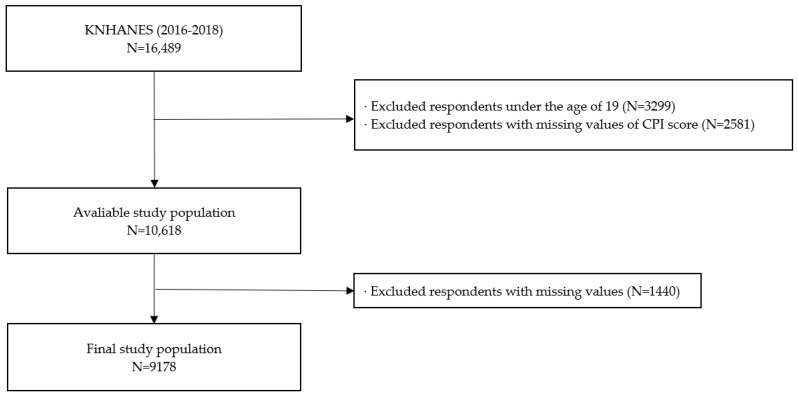
Flowchart of the study participants displaying the inclusion and exclusion criteria.; CPI, Community Periodontal Index; KNHANES, Korea National Health and Nutrition Examination Survey.

**Table 1 ijerph-20-04423-t001:** General characteristics of the study population.

Variables		Community Periodontal Index (CPI)													
		Male							Female						
		Total		No		Yes		*p*-Value	Total		No		Yes		*p*-Value
		*N*	%	*N*	%	*N*	%		*N*	%	*N*	%	*N*	%	
Total (*N* = 9178)		4161	45.3	1119	26.9	3042	73.1		5017	54.7	1874	37.4	3143	62.6	
Smoking Behavior								<0.0001							0.0130
	Non-smoker	1054	25.3	376	35.7	678	64.3		4391	87.5	1668	38.0	2723	62.0	
	Ex-smoker	1623	39.0	419	25.8	1204	74.2		344	6.9	123	35.8	221	64.2	
	Current smoker	1484	35.7	324	21.8	1160	78.2		282	5.6	83	29.4	199	70.6	
Age								<0.0001							<0.0001
	20–29	645	15.5	276	42.8	369	57.2		688	13.7	361	52.5	327	47.5	
	30–39	800	19.2	250	31.3	550	68.8		1007	20.1	453	45.0	554	55.0	
	40–49	936	22.5	238	25.4	698	74.6		1215	24.2	447	36.8	768	63.2	
	50–59	810	19.5	175	21.6	635	78.4		1044	20.8	319	30.6	725	69.4	
	60–69	585	14.1	114	19.5	471	80.5		718	14.3	202	28.1	516	71.9	
	≥70	385	9.3	66	17.1	319	82.9		345	6.9	92	26.7	253	73.3	
Marital status								<0.0001							<0.0001
	Married	2916	70.1	683	23.4	2233	76.6		3507	69.9	1234	35.2	2273	64.8	
	Divorced	146	3.5	30	20.5	116	79.5		302	6.0	80	26.5	222	73.5	
	Single	1099	26.4	406	36.9	693	63.1		1208	24.1	560	46.4	648	53.6	
Educational level								<0.0001							<0.0001
	Middle school	673	16.2	85	12.6	588	87.4		1057	21.1	250	23.7	807	76.3	
	High school	1498	36.0	416	27.8	1082	72.2		1767	35.2	652	36.9	1115	63.1	
	College	1990	47.8	618	31.1	1372	68.9		2193	43.7	972	44.3	1221	55.7	
Household income								<0.0001							<0.0001
	Low	477	11.5	93	19.5	384	80.5		632	12.6	179	28.3	453	71.7	
	Mid-low	939	22.6	230	24.5	709	75.5		1239	24.7	449	36.2	790	63.8	
	Mid-high	1280	30.8	345	27.0	935	73.0		1526	30.4	592	38.8	934	61.2	
	High	1465	35.2	451	30.8	1014	69.2		1620	32.3	654	40.4	966	59.6	
Region								<0.0001							<0.0001
	Metropolitan	1973	47.4	533	27.0	1440	73.0		2437	48.6	927	38.0	1510	62.0	
	Urban	1556	37.4	479	30.8	1077	69.2		1891	37.7	798	42.2	1093	57.8	
	Rural	632	15.2	107	16.9	525	83.1		689	13.7	149	21.6	540	78.4	
Perceived stress level								0.2388							0.1054
	Low	636	15.3	150	23.6	486	76.4		663	13.2	234	35.3	429	64.7	
	Mid-low	2433	58.5	667	27.4	1766	72.6		2858	57.0	1059	37.1	1799	62.9	
	Mid-high	934	22.4	259	27.7	675	72.3		1214	24.2	485	40.0	729	60.0	
	High	158	3.8	43	27.2	115	72.8		282	5.6	96	34.0	186	66.0	
Occupational categories								<0.0001							<0.0001
	White	1433	34.4	463	32.3	970	67.7		1419	28.3	633	44.6	786	55.4	
	Pink	495	11.9	128	25.9	367	74.1		846	16.9	284	33.6	562	66.4	
	Blue	1309	31.5	229	17.5	1080	82.5		657	13.1	174	26.5	483	73.5	
	Inoccupation	924	22.2	299	32.4	625	67.6		2095	41.8	783	37.4	1312	62.6	
Current drinking status								<0.0001							0.7788
	Never	1019	24.5	274	26.9	745	73.1		2457	49.0	914	37.2	1543	62.8	
	Monthly	1640	39.4	515	31.4	1125	68.6		1847	36.8	700	37.9	1147	62.1	
	Weekly	1502	36.1	330	22.0	1172	78.0		713	14.2	260	36.5	453	63.5	
Physical activity								<0.0001							0.0002
	Adequate	2077	49.9	614	29.6	1463	70.4		2210	44.1	888	40.2	1322	59.8	
	Inadequate	2084	50.1	505	24.2	1579	75.8		2807	55.9	986	35.1	1821	64.9	
Teeth Brushing Frequency								<0.0001							<0.0001
	≤1	464	11.2	102	22.0	362	78.0		164	3.3	43	26.2	121	73.8	
	2	1694	40.7	399	23.6	1295	76.4		1779	35.5	619	34.8	1160	65.2	
	≥3	2003	48.1	618	30.9	1385	69.1		3074	61.3	1212	39.4	1862	60.6	
Dental checkup status								<0.0001							<0.0001
	No	2616	62.9	609	23.3	2007	76.7		3016	60.1	1004	33.3	2012	77.7	
	Yes	1545	37.1	510	33.0	1035	67.0		2001	39.9	870	24.0	1131	76.0	

**Table 2 ijerph-20-04423-t002:** Results of factors associated with smoking and community periodontal index.

Variables		Community Periodontal Index (CPI)			
		Male		Female	
		OR	95% CIs	OR	95% CIs
Smoking Behavior					
	Non-smoker	1.00		1.00	
	Ex-smoker	1.20	(0.95–1.50)	1.19	(0.89–1.60)
	Current smoker	1.78 *	(1.43–2.23)	1.44 *	(1.04–1.99)
Age					
	20–29	1.00		1.00	
	30–39	1.41	(0.99–2.00)	1.18	(0.91–1.54)
	40–49	1.77 *	(1.24–2.51)	1.63 *	(1.24–2.15)
	50–59	1.83 *	(1.24–2.69)	1.93 *	(1.44–2.57)
	60–69	2.36 *	(1.53–3.64)	1.79 *	(1.27–2.52)
	≥70	2.87 *	(1.70–4.83)	1.88 *	(1.20–2.96)
Marital status					
	Married	1.00		1.00	
	Divorced	0.77	(0.51–1.17)	1.36	(0.97–1.92)
	Single	0.86	(0.64–1.14)	0.79 *	(0.63–0.98)
Educational level					
	Middle school	1.63 *	(1.15–2.23)	1.59 *	(1.18–2.14)
	High school	1.10	(0.89–1.35)	1.10	(0.92–1.31)
	College	1.00		1.00	
Household income					
	Low	1.47 *	(1.01–2.15)	1.03	(0.78–1.37)
	Mid-low	1.15	(0.90–1.46)	0.99	(0.82–1.21)
	Mid-high	1.05	(0.86–1.28)	0.93	(0.78–1.12)
	High	1.00		1.00	
Region					
	Metropolitan	1.00		1.00	
	Urban	0.77	(0.59–1.02)	0.73	(0.57–0.92)
	Rural	1.39	(0.91–2.10)	1.78 *	(1.19–2.67)
Perceived stress level					
	Low	1.00		1.00	
	Mid-low	0.89	(0.69–1.14)	1.02	(0.83–1.26)
	Mid-high	0.93	(0.68–1.27)	0.98	(0.77–1.25)
	High	0.94	(0.59–1.50)	1.25	(0.881.77)
Occupational categories					
	White	1.00		1.00	
	Pink	1.13	(0.85–1.52)	1.17	(0.92–1.48)
	Blue	1.48 *	(1.17–1.89)	1.06	(0.80–1.41)
	Inoccupation	0.72 *	(0.57–0.91)	1.00	(0.83–1.20)
Current drinking status					
	Never	1.00		1.00	
	Monthly	0.92	(0.74–1.14)	1.24 *	(1.07–1.43)
	Weekly	1.21	(0.95–1.55)	1.17	(0.95–1.45)
Physical activity					
	Adequate	1.00		1.00	
	Inadequate	1.03	(0.87–1.22)	1.04	(0.90–1.21)
Teeth Brushing Frequency					
	≤1	1.17	(0.87–1.57)	1.57 *	(1.03–2.39)
	2	1.21 *	(1.02–1.43)	1.02	(0.88–1.17)
	≥3	1.00		1.00	
Dental checkup status					
	No	1.62 *	(1.36–1.93)	1.60 *	(1.39–1.84)
	Yes	1.00		1.00	

* *p*-value < 0.05.

**Table 3 ijerph-20-04423-t003:** Results of subgroup analysis stratified by independent variables.

Variables	Community Periodontal Index (CPI)									
	Male					Female				
	Non	Ex-Smoker		Current Smoker		Non	Ex-Smoker		Current Smoker	
	OR	OR	95% CIs	OR	95% CIs	OR	OR	95% CIs	OR	95% CIs
Age										
20–29	1.00	1.10	(0.63–1.92)	1.82 *	(1.16–2.86)	1.00	0.83	(0.45–1.55)	0.87	(0.41–1.83)
30–39	1.00	1.40	(0.85–2.31)	1.78 *	(1.16–2.73)	1.00	1.23	(0.76–2.00)	2.73 *	(1.45–5.17)
40–49	1.00	1.31	(0.84–2.04)	2.21 *	(1.37–3.29)	1.00	1.34	(0.74–2.42)	1.37	(0.72–2.61)
50–59	1.00	0.88	(0.49–1.57)	1.34	(0.72–2.48)	1.00	1.88	(0.72–4.92)	1.62	(0.64–4.10)
60–69	1.00	1.38	(0.75–2.54)	2.57 *	(1.10–6.03)	1.00	0.98	(0.32–1.71)	1.76	(0.36–8.70)
≥70	1.00	1.59	(0.68–3.71)	2.71	(0.68–10.71)	1.00	3.56	(0.45–28.27)	-	-
Marital status										
Married	1.00	1.26	(0.96–1.65)	1.91 *	(1.44–2.53)	1.00	1.18	(0.81–1.72)	1.51	(0.81–1.72)
Divorced	1.00	0.07 *	(0.01–0.89)	0.06	(0.00–1.32)	1.00	0.90	(0.34–2.35)	4.44 *	(1.49–13.27)
Single	1.00	1.06	(0.65–1.72)	1.70 *	(1.18–2.43)	1.00	1.14	(0.70–1.86)	1.07	(0.61–1.87)
Educational level										
Middle school	1.00	1.70	(0.87–3.30)	3.15 *	(1.37–7.21)	1.00	1.12	(0.43–2.90)	11.40 *	(2.93–44.45)
High school	1.00	1.21	(0.83–1.77)	1.67 *	(1.14–2.44)	1.00	1.14	(0.68–1.93)	1.25	(0.77–2.03)
College	1.00	1.14	(0.85–1.52)	1.76 *	(1.34–2.32)	1.00	1.19	(0.80–1.77)	1.13	(0.66–1.93)
Household income										
Low	1.00	0.78	(0.37–1.67)	1.95	(0.84–4.55)	1.00	1.28	(0.44–3.72)	2.73	(0.95–7.87)
Mid-low	1.00	1.38 *	(1.47–4.45)	2.56 *	(1.47–4.47)	1.00	1.45	(0.83–2.54)	1.61	(0.85–3.05)
Mid-high	1.00	1.17	(0.78–1.74)	1.82 *	(1.22–2.71)	1.00	1.16	(0.72–1.88)	1.27	(0.72–2.27)
High	1.00	1.19	(0.85–1.68)	1.51 *	(1.09–2.11)	1.00	1.04	(0.61–1.77)	1.14	(0.60–2.17)
Region										
Metropolitan	1.00	1.39 *	(1.00–1.95)	2.16 *	(1.57–2.95)	1.00	1.19	(0.80–1.77)	0.95	(0.59–1.55)
Urban	1.00	1.22	(0.88–1.70)	1.75 *	(1.24–2.47)	1.00	1.18	(0.76–1.86)	2.24 *	(1.32–3.80)
Rural	1.00	0.43	(0.19–1.00)	0.74	(0.35–1.59)	1.00	1.19	(0.32–4.40)	1.48	(0.67–3.29)
Perceived stress level										
Low	1.00	1.41	(0.82–2.42)	4.02 *	(1.96–8.27)	1.00	1.52	(0.59–3.91)	1.96	(0.63–6.12)
Mid-low	1.00	1.16	(0.87–1.54)	1.72 *	(1.29–2.31)	1.00	1.17	(0.78–1.76)	1.48	(0.93–2.36)
Mid-high	1.00	1.28	(0.79–2.06)	1.87 *	(1.18–2.96)	1.00	1.18	(0.64–2.19)	1.25	(0.70–2.23)
High	1.00	0.55	(0.13–2.37)	0.75	(0.23–2.43)	1.00	0.93	(0.40–2.16)	1.29	(0.51–3.27)
Occupational categories										
White	1.00	1.19	(0.86–1.63)	1.68 *	(1.21–2.33)	1.00	1.21	(0.76–1.93)	1.23	(0.64–2.38)
Pink	1.00	0.72	(0.33–1.54)	1.29	(0.71–2.36)	1.00	0.95	(0.46–1.97)	1.57	(0.79–3.12)
Blue	1.00	1.08	(0.71–1.65)	1.80 *	(1.16–2.80)	1.00	0.65	(0.23–1.82)	2.92 *	(1.07–1.82)
Inoccupation	1.00	1.57	(0.99–2.48)	2.10 *	(1.28–3.43)	1.00	1.44	(0.91–2.28)	1.26	(0.75–2.28)
Current drinking status										
Never	1.00	1.48	(0.98–2.23)	1.90 *	(1.23–2.94)	1.00	1.22	(0.74–2.01)	1.84 *	(1.01–3.38)
Monthly	1.00	1.27	(0.92–1.76)	1.88 *	(1.37–2.57)	1.00	1.32	(0.87–1.99)	1.12	(0.67–1.89)
Weekly	1.00	0.87	(0.56–1.36)	1.45	(0.95–2.21)	1.00	1.09	(0.59–2.11)	1.59	(0.95–2.66)
Physical activity										
Adequate	1.00	1.12	(0.83–1.50)	1.83 *	(1.35–2.50)	1.00	0.97	(0.66–1.43)	1.11	(0.68–1.81)
Inadequate	1.00	1.32	(0.96–1.82)	1.77 *	(1.30–2.42)	1.00	1.43	(0.96–2.21)	1.88 *	(1.26–2.79)
Teeth Brushing Frequency										
≤1	1.00	1.12	(0.52–2.39)	1.32	(0.62–2.84)	1.00	4.59	(0.64–32.98)	3.18	(0.59–17.18)
2	1.00	1.52 *	(1.04–2.22)	1.41 *	(1.48–3.05)	1.00	1.26	(0.83–1.92)	1.21	(0.71–2.05)
≥3	1.00	1.05	(0.77–1.42)	1.70 *	(1.26–2.29)	1.00	1.16	(0.81–1.65)	1.65 *	(1.08–2.51)
Dental checkup status										
No	1.00	1.06	(0.79–1.43)	1.73 *	(1.29–2.33)	1.00	1.05	(0.74–1.48)	1.95 *	(1.29–2.95)
Yes	1.00	1.39 *	(1.01–1.91)	1.90 *	(1.40–2.59)	1.00	1.57 *	(1.02–2.42)	0.82	(0.48–1.38)

* *p*-value < 0.05.

**Table 4 ijerph-20-04423-t004:** Results of subgroup analysis stratified by pack years and smoking cessation.

Variables		Community Periodontal Index (CPI)			
		Male		Female	
		OR	95% CIs	OR	95% CIs
Pack-Years					
	Non-smoker	1.00		1.00	
	Pack Years < 5	1.11	(0.87–1.42)	1.16	(0.90–1.50)
	5 ≤ Pack Years < 10	1.75 *	(1.34–2.27)	1.71	(0.94–3.13)
	10 ≤ Pack Years < 15	1.47 *	(1.10–2.27)	1.47	(0.72–3.00)
	15 ≤ Pack Years < 20	1.80 *	(1.27–2.56)	1.96	(0.62–6.18)
	20 ≤ Pack Years	1.84 *	(1.38–2.47)	2.69	(0.90–8.05)
Smoking Cessation Status					
	Non-smoker	1.00		1.00	
	Ex-smoker (5 yr. > Cessation)	1.42 *	(1.04–1.96)	1.11 *	(1.71–1.74)
	Ex-smoker (5 yr. ≤ Cessation < 10 yr.)	1.69 *	(1.16–2.47)	1.07	(0.61–1.87)
	Ex-smoker (10 yr. ≤ Cessation)	0.93	(0.71–1.21)	1.32	(0.81–2.14)

* *p*-value < 0.05.

## Data Availability

The dataset used in this study is publicly accessible (https://knhanes.kdca.go.kr/knhanes/sub03/sub03_02_05.do (accessed on 1 January 2023)).

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
