# Peer review of "Association between Smoking and Periodontal Disease in South Korean Adults"

_ijerph, 2023, doi:10.3390/ijerph20054423_

Round 1
Reviewer 1 Report
Well constructed study and well written manuscript. It is good if you can improve the introduction more to explain the relationship of smoking with general and oral health. In the methodology section, it is good to provide details of periodontal measurements and the ways and people who did it.
Author Response
Please check the attachment below.
Best regards,
KAYUN SIM

Reviewer 2 Report
Dear authors,
The aim of the present cross-sectional study was to evaluate the association between smoking and the risk of having periodontal diseases in South Korean adults. I appreciate the authors' efforts in this paper with relatively large sample size. The manuscript is generally well-written. However, my main concern is that the justification for the paper is not clear for me. Numerous previous studies have been conducted on this subject and I don't understand what your article adds to the existing literature. Additionally, here are further details:
1) The aim of the study has to be mentioned in the abstract.
2) It is not clear that the authors are part of the KNHANES project group or just used the data obtained in that project.
3) Several sentences need to be revised:
"smoking is the main risk factor influencing treatment and healing." of what?
"that smokers have a stronger relationship with periodontal disease risk than ex-smokers and non-smokers"
"had longer pack-years"
....
4) The discussion fails to connect the findings to the relevant literature.
5) manuscript requires English revision.
Author Response

(The authors gave the same response as above.)

Reviewer 3 Report
This study aimed to evaluate the relationship between smoking and periodontal disease risk using health examination data and representative sample data from Korean adults.
The idea is interesting, but the implementation has some things that could be improved.
1. Please present the results numerically in the abstract.
2. In the introduction, explain in more detail the advantages of this study compared to similar ones. What are the gaps in the current research in this area?
3. What were the inclusion and exclusion criteria for the respondents?
4. There are no p and beta values in tables 2 to 5, so we do not know which variables are significant predictors.
5. In the discussion, you state according to the results in this and that table. State precisely the information you are discussing.
Author Response

(The authors gave the same response as above.)

Round 2
Reviewer 3 Report
I thank the authors for correcting the manuscript according to the recommendations and congratulate them.